# One of the Primary Functions of Tissue-Resident Pluripotent Pericytes Cells May Be to Regulate Normal Organ Growth and Maturation: Implications for Attempts to Repair Tissues Later in Life

**DOI:** 10.3390/ijms23105496

**Published:** 2022-05-14

**Authors:** David A. Hart

**Affiliations:** 1Department of Surgery and Faculty of Kinesiology, University of Calgary, Calgary, AB T2N 4N1, Canada; hartd@ucalgary.ca; 2McCaig Institute for Bone & Joint Health, University of Calgary, Calgary, AB T2N 4N1, Canada

**Keywords:** M, organ-specific cells, organ growth, growth and maturation, tissue-specific pericytes, pluripotent mesenchymal regulatory cells

## Abstract

Adult mesenchymal stem cells were reported more than 30 years ago. Since then, their potential to repair and regenerate damaged or diseased tissues has been studied intensively in both preclinical models and human trials. Most of the need for such tissue repair/regeneration is in older populations, so much of the effort has been performed with autologous cells in older patients. However, success has been difficult to achieve. In the literature, it has been noted that such progenitor cells from younger individuals often behave with more vigorous activity and are functionally enhanced compared to those from older individuals or animals. In addition, cells with the characteristics of mesenchymal stem cells or pluripotent mesenchymal regulatory cells exist in nearly all tissues and organs as pericytes since fetal life. Such evidence raises the possibility that one of the primary roles of these organ-specific cells is to regulate organ growth and maturation, and then subsequently play a role in the maintenance of organ integrity. This review will discuss the evidence to support this concept and the implications of such a concept regarding the use of these progenitor cells for the repair and regeneration of tissues damaged by injury or disease later in life. For the latter, it may be necessary to return the organ-specific progenitor cells to the functional state that contributed to their effectiveness during growth and maturation rather than attempting to use them after alterations imposed during the aging process have been established and their function compromised.

## 1. Background

Cells fitting the description of Mesenchymal Stem/Signaling Cells (MSC) have been isolated from a large number of adult tissues [1]. The original characteristics of MSC as defined by Caplan’s group [2] were adherence to plastic, expression of a subset of cell surface antigens, and the cells could be induced in vitro to differentiate towards different cell lineages, including chondrocytes, bone cells, and adipocytes. It has been noted that MSC from most tissues or fluids are very heterogeneous [3,4,5,6], and some sources appear to have unique features, including unique lectin-binding phenotypes [7]. Interestingly, MSC from bone marrow appear to preferentially respond to osteogenic stimuli [8], while MSC from synovium respond well to chondrogenic stimuli [8]. Thus, different locations may reflect the needs of different environments.

Of additional interest have been the reports that MSC from various sources of young animals differ from those of older animals [9,10,11,12,13]. The cells have been reported to differ in proliferation rate, transcriptome, Extracellular Vesicle (EV) content and secretion pattern. Cells from older animals appear to have undergone epigenetic modification compared to younger cells [14,15]. These attributes of MSC from younger individuals have led some to cryopreserve their cord blood or Wharton’s jelly MSC in case they are needed later in life [16,17].

Therefore, as the most significant need for tissue repair or regeneration due to accidents, disease, or other causes is mainly for the elderly, the cells appear to be somewhat compromised when there is the greatest need to take advantage of their abilities. This may be one of the reasons that injection of autologous MSC from older individuals to elicit repair or regeneration of damaged or injured has not been met with consistent success. However, some successes using in vitro generated tissue-engineered constructs derived from synovial MSC in vivo for the repair of cartilage defects [18,19] in humans have been noted; there have also been a number of less successful endeavors with other tissues [20].

This lack of success in achieving the expectations that accompanied the initial reports of MSC as “stem cells”, led Caplan [21,22] to suggest that perhaps MSC should not refer to stem cells but to the designation of them as Medicinal Signaling Cells and their primary role was in tissues in their pericellular location to foster the integrity of cells in an organ via release of chemical mediators and exosomes containing mediators and other important regulatory molecules such as miRNA [23,24]. The importance of these exosomes or EV-containing mediators and complex contents is an emerging field and one that indicates the significant potential to impact the repair of damaged tissues [25,26,27,28]. It should be noted that most of the effort with MSC has been focused on the adult scenario and the potential of the MSC to affect the response to injury [29].

While this concept of MSC being a Medicinal Signaling Cell is intriguing, one cannot dismiss the option that there are multiple subsets of MSC that serve different functions [30,31] in different locations at different timings in the growth, maturation, and senescence of a host, such as a human. One should not dismiss the pluripotency of these cells from the name [3]. This conundrum has led to the suggestion that perhaps what have been called MSC should be renamed Pluripotent Mesenchymal Regulatory Cells (PMRC) to reflect their in vivo function associated with being organ-specific pericytes and their in vitro pluripotency [32]. The current review is focused on the possibility that in vivo, the pericyte function and the pluripotency of these cells are integrated during growth and maturation to facilitate the coordinated growth of organs and the integration of their cellular complement to yield a functioning system that can respond to systemic modulators of growth and maturation. Thus, the pericytes with the attributes of PMRC are integrated into the functioning of a growing organ via differentiation in an organ-specific manner in response to the local environment (liver cells, heart, muscle, kidney, etc., ECM, endothelial cells of the microvasculature, innervation).

## 2. Introduction

During in utero development, the various organs form via developmental programing. Subsequently, in humans, the organs contain the necessary cells and cell types to yield an integrated group of cells that will contribute to a functioning system via cell–cell communication, an organ-specific extracellular matrix (ECM), and further differentiation of the associated parenchymal cells, endothelial cells of the microvasculature, endogenous neural elements, and endogenous pluripotent cells known as “mesenchymal stem cells or medicinal signaling cells”. Thus, fetal development lays the groundwork for the organization of an organ and the functional integration of its component cells. Therefore, within a specific organ, the endothelial cells of the microvasculature, the various parenchymal cells, the MSC/pericytes, the innervation where appropriate, and the ECM must function internally as a functional unit and externally as a unit functionally integrated into the context of the systems biology of the host.

A critical component of this system is the endothelial cells of the microvasculature. As the microvasculature is the key interface between the host and specific parenchymal cells and pericytes within an organ, they play a unique role in regulation. As reviewed recently by Augustin and Koh [33], the endothelial cells of the microvasculature undergo organotypic differentiation to accommodate specific organ systems. How this develops and is maintained is still undefined for the most part, but it exemplifies the complexity of the cell–cell and cell-matrix interactions that contribute to a functionally integrated organ system.

With the likely exception of the liver [34], overt damage to a human organ or removal of part of it will not lead to tissue regeneration, while some species such as planarians [35], some amphibians [36] and fish [37] have retained such abilities. Thus, in humans, this natural regenerative process is quite limited [38].

Once the template for an organ or a tissue is developed during fetal progression, the organs and tissues of a human continue to mature and grow in utero, particularly during the third trimester. Thus, at the time of full-term birth, the human is able to function in a mostly coordinated and integrated manner with the assistance of the mother via lactation. In this context, an organ such as a liver, kidney, heart, lung, or centers of the brain is/are set to continue to grow and mature in the postnatal environment.

In contrast, some tissues of the musculoskeletal system, such as ligaments and tendons designed to function in a mechanically active environment, exist as cellular tissues devoid of much ECM at the time of birth (i.e., “cell-rich and matrix-poor”). Subsequently, the tissues will progressively lay down a more organized ECM between the cells yielding a tissue that is hypocellular at skeletal maturity but is rich in ECM. This process is driven in part by the demands of the biomechanical environment and the presence of anabolic factors. Removing the mechanical stimuli from such an in vivo growing ligament stops growth and maturation [39]. In this case, the tissue is no longer responsive to any in vivo anabolic stimuli.

Thus, as the growth and maturation of organs in the early postnatal period (0–3 years of age) occur in a regulated manner consistent with an integrated systems biology approach, the dependence on anabolic signals is likely more common. Whether much of this growth occurs as a continuum or a series of regulated “steps” (i.e., growth “spurts”), or both may depend on the specific tissue, but at skeletal maturity, the multicellular organs are comprised of different cell types, function as integrated units. However, it is clear that many organs exist as a “functional unit” consisting of various cell types [endothelial cells, pericytes (some of which exhibit pluripotency properties and can be called mesenchymal stem cells or medicinal signaling cells or pluripotent mesenchymal regulatory cells; 2, 32), organ-specific cells, neural elements] (Figure 1), and continue to expand during the growth and maturation phase of the lifespan after birth via controlled and coordinated proliferation of various cells with orderly deposition of ECM, controlled neovascularization and accompanying innervation, as well as endogenous MSC/pericytes.

As outlined in Figure 1, as a result of developmental programs individual organs/tissues can be defined as a “functional unit” consisting of one or more organ-specific cells that define the function of the organ (i.e., osteoclasts and osteoblasts and osteocytes in bone; liver parenchymal cells, etc.). Different organs also appear to contain other what could be considered “regulatory” cell types, such as organ-specific differentiated endothelial cells [33,40], neural elements that can influence both the microvasculature in various situations [41,42] as well as other organ-specific cells (i.e., parenchymal cells in the liver; osteoblasts and osteocytes in bone as examples), and pericytes which often exhibit the characteristics of what have been called mesenchymal stem cells [43].

However, not all of the pericytes may be what has been called MSC, but their heterogeneity may also represent other cell types [44]. The role of the neural elements is not clear as following transplantation of an organ such as a liver, the nerves have been transected and there does not appear to be an overt loss of function [45]. Analogous to the endothelial cells, such organ-localized pluripotent regulatory cells could take advantage of the lineage “plasticity” and ability to differentiate in vitro towards different lineages [2,46], and differentiate locally to enhance their “signalling” role to optimize the contents of their extracellular vesicles and their secretions to maintain the integrity of the specific “functional units” that constitute an organ system. Such differentiation could be directed by signals from the endogenous organ-specific cells or in part, from the cells of the microvasculature. The latter could be influenced by the resident pericytes to maintain differentiation as outlined in the proposed scheme in Figure 1. In this proposed scheme, the tissue-specific pericytes would be ideally located to facilitate regulation in response to systemic mediators in the vasculature and translate them into a more organ-specific response pattern. Thus, the tissue-specific pericytes function as facilitators, amplifiers, and regulators for fine-tuning response patterns. There may also “cross-talk” between the regulatory pericytes and the endothelial cells of the microvasculature [47]. In the context of such a role, there would be more to it than maintenance of integrity, but also controlled expansion during the growth and maturation stage of life (0 to ~10/12 years of age) and then further sex-specific maturation following the onset of puberty. Such regulation by these tissue-specific regulatory cells could also be influenced in females by the conditions of pregnancy in an organ-specific manner.

The last element in the functional unit of an organ that should be mentioned is the extracellular matrix (ECM). As discussed recently by Yang et al. [48], it is likely that each organ system has a unique ECM that contains specific components or perhaps even unique splice variants of common molecules. This ECM can arise from the pericytes [49] or the parenchymal cells of the tissue. Specificity may be manifested via the glycolytic linkages in glycoproteins or on cells [7,50]. Such an organ-specific ECM could contribute to the localization of the cells in the organ via cell receptors as has been reviewed recently by Hart [51], as a source of biologically active molecules bound to the elements of the ECM, such as in bone [52], as an example, and contribute to the morphology of specific organs [53]. Furthermore, the ECM in an organ is likely dynamic and can be altered by injury or during the aging process [54,55], factors that could impact organ integrity and functioning.

Growth and maturation of tissue and organs progress at very individualized rates until the next transition in maturation occurs, that of the onset of puberty. While some sex-dependent aspects of growth and maturation occur prior to puberty, the onset of puberty is associated with another burst of growth and sexual development/maturation in both males and females. However, the onset of puberty, particularly in females, sets the stage for subsequent pregnancies and the altered regulation of many tissues and organs to accommodate the necessary adaptations leading to a successful pregnancy. As the onset of puberty is also accompanied by a time of growth as well, this requires the successful coordination and integration of both aspects of growth and the adaptations associated with puberty. Thus, in the time frame from the onset of puberty to skeletal maturity, there must be retention of integrated organ function that occurred during pre-pubertal growth and maturation alongside continued growth but with an altered maturation goal after puberty onset. In this context, the endogenous pericytes in each organ system or tissue must continue to serve unique functions that are integrated with the associated cell types and regulatory elements. With increasing commitment, many, if not most, likely develop epigenetic [56] or carbohydrate [7] signatures to reflect this commitment. Of note, puberty itself is accompanied by epigenetic alterations to many cells [57,58,59,60], so this could also include cells such as those that are labelled MSC/pericytes in different tissues.

For females, the next opportunity for regulatory impact on organ systems and tissues is pregnancy and lactation, potentially multiple times throughout most of evolutionary history. Obviously, many systems have to adapt to the conditions of pregnancy, including organs and systems such as the cardiovascular system, kidneys, lungs, the MSK system and others, with the adaptations somewhat dynamic throughout the nine-month pregnancy as the fetus matures, gains weight and puts different stresses on the maternal systems. After birth and during lactation, there are other adaptations required. However, many of the systems affected by pregnancy return to near normal conditions [but not all of the MSK systems may return to pre-pregnancy conditions]. Therefore, the regulation of these systems must be at least in part reversible. In the return to the post-pregnancy environment, functional integrity is maintained. The interactions between cells in an organ system must be resilient against loss of biological integrity, including the role of the MSC in the various organs.

With aging, the functioning of a number of systems and organs can decline in different populations, and again males and females are affected differently. Some of this decline in function may relate to genetics and epigenetic changes occurring during life [61,62,63]. In females, this relates to the onset of menopause at age ~45–50. Interestingly, the average lifespan for much of evolutionary history was likely <30 years age, so the number of females who actually went through menopause until the relatively recent past was only a subset of females who reached 50 years of age.

Of note, menopause appears to be a process that can take years and thus is not an acute event. Therefore, the changes occurring in a variety of tissues and organs following the decline in ovarian function and systemic levels of sex hormones and development of secondary effects is not an abrupt change but rather a relatively slow process that in different subsets of females can result in conditions such as osteoporosis, obesity, increased cardiovascular risk, and risk for cognitive decline and the onset of dementia [64]. With regard to osteoporosis, some females lose bone integrity in a somewhat wide range of rates, with some losing much more bone than others [65,66] and many similarly aged females not losing much at all. Therefore, there is likely some genetic basis for the rate of bone loss after menopause in those at risk for osteoporosis development. This is likely true for osteoporosis, dementia and obesity, while that for cardiovascular disease risk is still not clear. Thus, this focus on specific targets in these post-menopausal conditions tends to shape the research effort and may contribute to the lack of progress in some areas [64].

## 3. Potential Role of Pluripotent Organ-Specific Pericytes in Growth and Maturation, as Well as Senescence and Decline in Organ and Tissue Integrity

As mentioned above, progenitor cells have been found in just about every tissue and organ system that has been examined [5], and these cells are usually found as pericytes in close proximity to the microvasculature as well [43,67,68]. Thus, these tissue-localized MSC are well situated to respond to biological signals traveling through the blood, as well as signals from the tissue-specific endothelial cells that are in direct contact with the bloodborne signals. Furthermore, signals from the MSC-like pericytes could engage in crosstalk with the endothelial cells for mutual benefit [47]. In addition, they are also in a position to be influenced by neural elements, directly or indirectly via neural influences on the organ-associated endothelial cells and or the tissue-specific pericytes. Interestingly, MSC have been reported to express neuropeptide receptors [69] such as CGRP [70] and neuropeptide Y [71,72].

Furthermore, when MSC are isolated from tissues or organs of young animals and adult or older animals, the younger cells exhibit characteristics that differ from the older MSC both in vitro and in vivo. This can be manifested at the level of the extracellular matrix the MSC provides [49], cell proliferation [73], susceptibility to oxidative stress [74], lineage differentiation [75], and functionality in vivo [76,77] and in vitro [73]. Thus, MSC from younger animals appear to be well suited to assist in the growth and maturation of tissues and organs when they are associated with specific tissues, such as pericytes. This concept is outlined in Figure 2.

A role for the tissue-specific pericytes in the growth and maturation of organs and tissues, as outlined in Figure 2, would be one of amplification of circulating systemic molecules that mediate such growth and maturation. In such a role as an organ-specific amplification component of the organ that is differentiated to optimize the delivery of specifically tailored extracellular vesicles and secretions, systemic growth mediator effects would be greatly amplified by these cells. However, as outlined in Figure 2, such a scheme does not imply that such systemic mediators of growth could not also directly impact the organ-specific parenchymal cells.

Therefore, some differentiation potential is also required of progenitor pericytes as they integrate into the functionality of an organ at the level of the parenchymal cells. It should also not be forgotten that the progenitor pericytes could also function to maintain the differentiation of the organ-specific endothelial cells of the microvasculature [78]. Thus, in this role, MSC could serve as sentinels and gatekeepers of regulatory signals and functions [79], but critically, during growth and maturation when the orderly growth of both the organ-specific parenchymal cells and the microvasculature is required. Of note, there appears to be an onset of MSC senescence in bone in late puberty in mice [80], potentially indicating that this could be due to the completion of a role during growth and maturation.

Being integrated into the unique environment of a tissue or organ at an early age, as well as being positioned in close proximity to the microvascular component, and capable of secreting relevant molecules as well as regulated “shedding” exosomes or extracellular vesicles containing relevant molecules that can migrate to target cells in a paracrine manner, make the cells currently called MSC ideal candidates to “translate” generalized anabolic factors in the bloodstream during growth and development into signals that are more organ- or tissue-relevant or impactful to a coordinated growth and maturation of each tissue or organ to retain function during periods of transition as in growth and development. Thus, these cells that have been called MSC may more appropriately be called Pluripotent Mesenchymal Regulatory Cells (PMRC) [32], a term that captures both the differentiation and signaling functions and abilities of MSC when becoming tissue-specific pericytes rather than using two different terms as suggested by Caplan [21,22]. In such a role, the pluripotent aspects of their capabilities may have benefits in maintaining tissue integrity via their self-renewal potential, as well as their ability to replace cells that have died via apoptosis or via errors in replication, and their ability to secrete tissue-specific mediators and release extracellular vesicles. Thus, the ability of these pluripotent cells to differentiate in order to optimize their regulatory functions in a tissue/organ-specific manner may be a critical feature of the cells. Additionally, an injection of BM-derived [81,82] or adipose tissue-derived [83,84] MSC into the knees of patients with osteoarthritis can often relieve the pain and inflammation in such joints; resident tissue-specific pericytes could also potentially function to dampen low levels of endogenous inflammation arising in such an environment to avoid evoking an inappropriate systemic inflammatory response and an elevated risk for loss of organ function due to fibrosis or cell death. Such immunomodulatory cells in bone marrow or adipose tissue may be influenced by donor age [73]. However, care must be taken when interpreting some characteristics of these cells, as some properties may be influenced by passaging the cells in vitro [85], possibly due to epigenetic drift [86].

## 4. What Is the Role of Pluripotent Progenitor Cells That Circulate or Are Detected in the Free State?

While the above designation of PMRC for cells now routinely labeled MSC (mesenchymal stem cells or medicinal signaling cells) that are associated with tissues and organs is likely appropriate, the question then arises regarding the role(s) of these cells that are circulating in the bloodstream [87] or found in fluids such as synovial fluid [8,88]. They could represent a separate subset of pluripotent progenitor cells, or they could reflect just another manner to exert regulatory functions. The bloodborne pluripotent cells likely arise from the bone marrow [BM] compartment, where they are believed to support hematopoiesis [89,90] and also participate in bone fracture healing [91] but may decline in effectiveness with age [92]. Interestingly, MSC from the periosteum may also participate in fracture healing [93]. Therefore, in the BM, they can exert a regulatory function and perhaps other functions related to the pluripotency (it has been shown that pluripotent progenitor cells from the BM preferentially differentiate towards the osteogenesis lineage; 8, 46). Thus, if the population of these cells circulating in the vascular system is likely BM-derived, they could then also exert some anti-inflammatory regulatory influences at distant sites.

The role of pluripotent progenitor cells found in the synovial fluid (SF) of articulating joints may be somewhat different from analogous cells circulating in the blood. The SF cells may be potentially unique to that environment as they exist as free-floating cells in the synovial fluid with their unique molecular composition. The SF-derived cells likely come from the synovium as they exhibit very similar characteristics [94]. However, it is also possible that some of the SF PMRC are derived from the infrapatellar fat pad (Hoffa’s fat pad) [95].

It has been noted that SF-derived progenitor cells exhibit multipotency [95] and appear to prefer to differentiate towards chondrogenesis and thus a more cartilage forming behavior [8]. This preference may also be exhibited by analogous cells derived from other joint tissues as well, as discussed by Huang et al. [96]. While articular cartilage has been reported to contain some pluripotent cells [97], it does not have discernable vascularity and is aneural and thus is devoid of some regulatory cells. While this deficiency may hamper the repair of injured articular cartilage, the MSC-like cells within the articular cartilage could serve other functions, such as during growth and maturation [97], where they could provide tissue-specific secretions to assist in the coordinate growth of the different layers of the cartilage.

Furthermore, other tissues, such as the menisci of the knee, also have parts that are devoid of microvascularity and innervation [98,99,100]. A joint such as a knee functions as an organ system [101,102] with a unique configuration involving the synovial fluid, the role of synovial fluid pluripotent progenitor cells is similar to those in more “connected” organs but is adapted to the unique requirements of a joint to accommodate the biomechanical needs and the lack of innervation and vascularity in some elements of the joint. That is, the synovial fluid pluripotent cells may function as a unique source of cells for the intra-articular joint tissues [8,46] but one that can be affected by joint diseases [88,103]. Thus, taking the view that these cells contribute to the regulation of normal tissues and organs during both growth and maturation as well as the maintenance of integrity as an adult, could allow for a unified theory as to their roles and function in a diverse set of conditions or environments involving both their differentiation potential and a role as a source of regulatory molecules.

## 5. Does the Decline in Tissue-Specific Pericyte Function with Age Have Implications for a Potential Role in Loss of Health?

From the literature, the number of pluripotent cells declines with aging [104]. In addition, such cells from older members of a species appear to exhibit less proliferative potential than those from younger individuals and also appear to have an altered gene expression phenotype [9,75]. Additional reports indicate an increased susceptibility of these cells to oxidative stress occurs with aging [74]. Some of the age-dependent differences are not species-dependent and appear to be a more general rule. Thus, with aging can come declining numbers of these pluripotent cells, and also impaired function [76,105]. How a loss of function with age is manifested is unknown, but one possibility is via epigenetic modification [14,15,75,106,107]. As some epigenetic modifications can occur as a result of life experiences and in the case of females, a transition such as menopause, the changes may be at a personal level and not a species level. If these changes in the function of the pluripotent cell populations are extended to the post-menopausal years of females, the withdrawal of the regulatory influence of sex hormones on the tissue-specific pericytes could contribute to aspects of some of the post-menopausal diseases that occur in different subsets of females.

With the potential loss of regulatory cells such as tissue-specific pericytes in terms of numbers or functional level (i.e., via epigenetic alterations) in various tissues during aging could have implications for the development of compromised tissues or organs and increased risk for disease development as a natural consequence of a deficient response to an exogenous insult or due to inability to control endogenous alterations leading to infiltration by inflammatory cells and more overt loss of function. In connective tissues that have biomechanical stimulation as an important factor in maintaining the integrity of the tissues, the decline in stimulation due to sedentary behavior could also contribute to tissue-specific pericyte dysfunction during aging. Interestingly, Gunawardene et al. [87] reported an association between circulating osteogenic progenitors and disability and frailty in older individuals.

An additional factor that may affect the functionality of pluripotent cells across the lifespan, but particularly in older populations who may require more pharmacological support, is taking drugs that could affect the functionality of these cells. High-dose glucocorticoids [108] and some antidepressants [109] have been reported to affect MSC.

Thus, even if a primary role for tissue-specific pericytes is during growth and maturation, a continued role in the maintenance of organ/tissue integrity after skeletal maturity may be likely, particularly as it is related to the ability to repair tissues from modest physical injury or following exposure to environmental chemicals resulting in injury or even death of parenchymal cells (Figure 3). In this scenario, EV or secretions from these tissue-specific regulatory cells could initiate the repair of injured cells and also block the effectiveness of pro-inflammatory signals from the injured cells via anti-inflammatory signals from the tissue-specific pericytes. In the event of limited parenchymal cell death, through controlled cell division and differentiation, these pericytes could also replace these damaged cells. However, an age-related decline in the ability of such cells to continue to maintain integrity could pose a risk for pathology development and loss of organ function. Additionally, with such a scenario, there is also the possibility that there may be a genetic variation that contributes to the longevity of functional tissue-specific pericytes in specific organs and tissues, a topic that could yield further insights through focused research efforts.

Thus, much evidence indicates that progenitor cells from a variety of locations undergo a loss of numbers and function with aging. Therefore, once skeletal and biological maturity is gained, such cells in specific organs or tissues likely perform more of a maintenance role in these locations. However, with advancing aging, the remaining cells appear to lose function, a set of properties that may compromise their effectiveness in the repair and regeneration of tissues injured or diseased. Therefore, the use of autologous progenitor cells from older individuals for such repair and regeneration purposes may not yield optimal results, and there is a need to revitalize such compromised cells or use allogeneic cells from younger donors to enhance the success of the interventions.

## 6. Is There Potential to Reverse the Decline in Tissue-Specific Pericytes Function with Aging?

From the above discussions, a role for tissue-specific pericytes in early growth and maturation has been made, as well as a brief discussion regarding the implications of the loss of the numbers and functioning of such cells on risk for loss of health. It is clear that much of the literature regarding the use of pluripotent cells for tissue engineering and other applications of free cells has been with older populations where tissues and organs have been damaged by the aging process and disease. Many of these issues have been reviewed recently by Hart [51]. Such approaches for the repair and regeneration of tissues/organs damaged by disease are complicated by the presence of an inflammation that can compromise the functionality of the cells [88,103] or the effectiveness of the effort [110]. However, some loss of these tissue-specific and more general pluripotent cells (i.e., in BM, SF, blood) occurs with aging, and no obvious disease association. Therefore, the answer to the question posed by this section depends in part on the mechanisms involved in the loss of these cells with aging. Thus, it could relate in part to the age-related loss of systemic molecules which support the health of such cells.

Alternatively, it could relate to the development of molecules that actually foster the loss of function and possible death of the cells. A number of decades ago (~1970s), literature developed regarding the existence of molecules called chalones, which are anti-proliferation molecules [111,112,113]. Such chalones could act directly on cells in tissues or indirectly affect growth via suppression of a central positive mediator [114]. An example of the latter is somatostatin, which has been called a “universal” chalone via suppression of growth hormone release [114]. The concept gradually fell out of favor but has undergone a resurgence [115,116]. However, the concept is interesting as a potential mechanism for the loss of specific cells or alterations in their functioning during aging as the ratio of positive mediators/chalones gradually favors the chalones.

While not a field that is well developed at the molecular level, there is some literature, primarily using rodent models, that is supportive of the concept that systemic factors are present in the blood of old and young animals that can influence cell activity both in vitro and in vivo. Conboy et al. [117] reported that aged progenitor cells could be rejuvenated by exposure to serum from young mice and also via parabiotic studies with young and old animals. More recently, Hu et al. [118] reported that extracellular vesicles from human umbilical cord blood could ameliorate bone loss in senile osteoporotic mice. Furthermore, Kiss et al. [119] recently reported that old blood from heterochronic parabionts accelerates vascular aging in young mice. Such literature supports the concept that the systemic environment in the young is very different from older individuals, and age-related circulating mediators can influence progenitors.

There are several implications of such findings. First, the factors in the serum or blood as studied have likely influenced the maintenance function of regulatory cells, such as the tissue-specific pericytes, and there is no evidence that could be found to support a unique role in organ growth. However, the findings support the concept that the changes accompanying aging with pluripotent cell populations are actually reversible. Secondly, one could conclude that it is the ratio of positive/negative regulators in the blood that is important, particularly in relation to the parabiotic studies. That is, blood from a young animal can affect the aging process even in the presence of aged blood and vice versa, so it is a competitive environment for effectiveness. Thirdly, in the use of pluripotent progenitor cells for tissue repair interventions, most often, the patients are older, and autologous cells are used for injection or tissue engineering purposes. However, cells defined as MSC from cord blood or umbilical cord tissue that is either allogenic or from cells stored from birth have been studied in both human and preclinical models [120,121]. From the above discussion, using autologous cells defined as PMRC or MSC from older patients could contribute to a lack of positive outcomes from the interventions, and perhaps transient exposure of the cells to molecules or EV from young blood may improve outcomes. This is perhaps an area for future research.

A key question that remains from the evidence described above is how such effects of old and young blood work at the molecular level. As reviewed by Spehar et al. [122] and others [9,123], considerable evidence indicates that stem cell functionality declines with aging, and an ability to restore functionality could have a number of implications. Therefore, it would be important to better understand how blood from young animals or humans actually can rejuvenate aged progenitor cells. While speculation at this point, one possibility is related to the reversal of age-related epigenetic alterations to progenitor cells in different tissues. During aging, a number of reports have indicated that progenitor cells become epigenetically altered [14,15,106,107], and such changes may reflect the loss of function with aging and impede the success of their use for tissue repair and regeneration. Thus, being able to reverse such aging-associated declines in function could be of great benefit when used in vitro to enhance tissue engineering approaches for tissue repair or even in vivo to reverse age-related decline in organ functions and risk for pathology.

## 7. How Realistic Are the Expectations of Using Pluripotent Progenitor Cells to Facilitate Tissue/Organ Regeneration?

As discussed above, more than 30 years of research investigating the application of MSC to facilitate tissue repair or regeneration using either injection of purified free cells or tissue-engineered constructs have yielded mixed results regarding “success”. While such research activity has generated considerable information regarding what MSC are and are not, the use of free cells injected systemically or into local sites of injury, particularly when the cells used are from a non-homologous site, has not achieved the expectations that existed in most cases. However, some tissue engineering applications of MSC have taken advantage of the proliferative and differentiation capacities of such cells to make inroads into the repair of some damaged tissues. One of these is articular cartilage, where implantation of tissue-engineered constructs (TEC) prepared with synovium-derived MSC into both large preclinical models [124,125] and humans [18,19] have led to success in regenerating the damaged cartilage. Thus, even if the natural role of tissue-specific pericytes is to mainly serve as a regulatory cell to maintain tissue and organ integrity during the lifecycle, it should still be possible to exploit the pluripotent potential of the cells for other purposes.

To further exploit the potential of pluripotent progenitor cells, it will be important to step back and address some challenging issues. First, if aging affects critical functionality aspects of the cells, what does this mean for using autologous cells from older people rather than cells from younger sources (i.e., cord blood, Wharton’s jelly, and placenta)? Based on the above discussion, cells from younger sources may be the better choice for several reasons. Secondly, if tissue-specific pericytes are regulatory cells integrated into the functionality of a tissue or organ, what are the implications of using readily available, but non-homologous sources like BM or adipose tissue for cells to be used in an organ or tissue that is quite different? Obviously, it would be important to better understand the limitations of such approaches and develop methods to overcome these limitations in a tissue/organ-specific manner. Thirdly, little is known regarding the recognition systems that these pluripotent progenitor cells use to localize and associate with the ECM and other cells in a tissue [51]. That is, how can one use endogenous recognition molecules to enhance localization to specific sites, or failing that, can one enhance localization and incorporation at a site via engineered surface molecules [51].

Thus, understanding the natural role of tissue-specific pluripotent pericytes across the lifespan is one goal of the research, while exploiting the pluripotent abilities of such cells is a separate goal. However, to achieve the expectations of the use of these cells will require the integration of the two goals, and they are likely inter-dependent.

## 8. Focus on the Pluripotency of the Cells Rather Than Their “Stemness”

When Caplan suggested the name Mesenchymal Stem Cells, it was based on the fact that cells with a set of characteristics (adherence to plastic, expression of a subset of cell surface molecules and not others) could be induced to differentiate into cells of the chondrogenic, osteogenic and adipogenic lineage by specific “cocktails” of growth factors, chemicals, media, and serum [2]. The was a focus on the “stem cell” aspects of these in vitro processes, and this led to a tsunami of research focused on the “stemness” of the cells for the next 30 years. When this effort did not lead to the expected success, Caplan [21] suggested a name change to Medicinal Signaling Cells with the abandonment of the term Stem Cells. However, this did not address the pluripotency of the cells or the fact that the lineage-specific differentiation was not important in and of itself other than to confirm that the cells had pluripotency abilities [32]. The present discussion would suggest that the pluripotency of the cells is critical for the ability of the cells in question to adapt and integrate into each organ system in an organ-specific manner, leading to the coordinated growth and maturation of each organ with the retention of their integrated functioning.

In this scenario, the pericytes with the characteristics of regulatory cells operate in the context of specific organ environments to facilitate the delivery of organ-specific anabolic signals during growth and maturation via translation of generalized endocrine factors and specific inputs into signals required for the differentiated proliferation and functioning of the endogenous cells of each organ [Figure 1]. This is likely maintained in a paracrine manner where the endogenous cells release unique mediators that instruct the pericytes to differentiate and remain in an appropriate manner that is optimized for that specific environment [Figure 1].

While it is unlikely that the pericytes with the characteristics of these tissue-specific progenitor cells are the only intermediaries in such tissue/organ regulation, they function in critical ways to maintain the integrity and optimize response patterns to stimuli when needed, such as during growth and maturation, and then subsequently to facilitate effective maintenance of the organ function. Thus, while most organs are likely regulated by both internal paracrine mechanisms, as well as external endocrine, metabolic and neural mediators and mechanisms, the functioning of the tissue-specific pericytes must be integrated into the response pattern of an organ even when they may not directly involve, such as with glucose levels and insulin responses involving beta cells of the pancreas, for example. Thus, organ regulation involves the tissue-specific pericytes, but it is not solely dependent on them and their functioning across the lifespan.

## 9. Conclusions

After more than 30 years since the initial reporting on pluripotent cells finding that they could be isolated from adult tissues, they are still being evaluated for their potential to facilitate tissue/organ repair and regeneration [2]. While considerable information has been gleaned from their study, success towards the goal of tissue/organ repair has been challenging. In retrospect, some of this challenge appears to have stemmed from the use of autologous progenitor cells from older individuals or animals in an attempt to either alter disease symptoms or repair damaged tissues or organs. While some successes were noted in the repair of some tissues [18,19,124,125], the outcomes were not always consistent. Partly due to the challenges faced, Caplan [21] then suggested changing the name from mesenchymal stem cells (MSC) to medicinal signaling cells (MSC) to indicate their potential primary role in the adult to be a source of biologic signals via secretions or release of extracellular vesicles [126]. As this definition did not capture the pluripotency of the cells, it was further suggested by Hart [32] that the cells be called pluripotent mesenchymal regulatory cells (PMRC) to account for both the pluripotency and signaling properties of the cells.

The finding that progenitor cells from young individuals exhibited many properties that appeared to be lost during the aging process has led to the concept that was presented that the primary role of tissue-specific pericytes is during growth and maturation, with a secondary role after biologic maturity being one of maintenance of tissue/organ integrity. In these capacities, these cells as organ-specific differentiated pericytes in each tissue could optimize the impact of systemic growth and maturation signals for their specific organ/tissue and then later assist in the maintenance of such integrity. This concept aligns well with the available data on such cells from young versus old donors, as well as the literature regarding the biological effects of blood from young animals on old cells and vice versa.

This concept also has implications for the use of progenitor cells to repair and regenerate damaged tissues and organs, a situation that is primarily the domain of older individuals. One major implication is that there is a need to both develop methods to reverse the age-related changes in such progenitor cells if autologous cells from older individuals are still to be used, potentially by reversing epigenetic alterations occurring during aging, and also identifying the components of blood from young individuals that can stimulate their effectiveness. Similarly, there is a need to identify components in blood from older individuals that appear to have the ability to compromise the functioning of cells from young individuals (i.e., chalones). Developing solutions for these issues will be critical to moving forward as we need to optimize the effectiveness of such cells to mimic growth and maturation, and not just the maintenance function of the cells as the individual ages.

Thus, further validation of the concept regarding the role of tissue-specific pericytes at different stages of the lifespan will be important for the understanding of the regulation of growth and maturation, and to further optimize the use of progenitor cells to repair and regenerate injured or damaged tissues and organs.

## Figures and Tables

**Figure 1 ijms-23-05496-f001:**
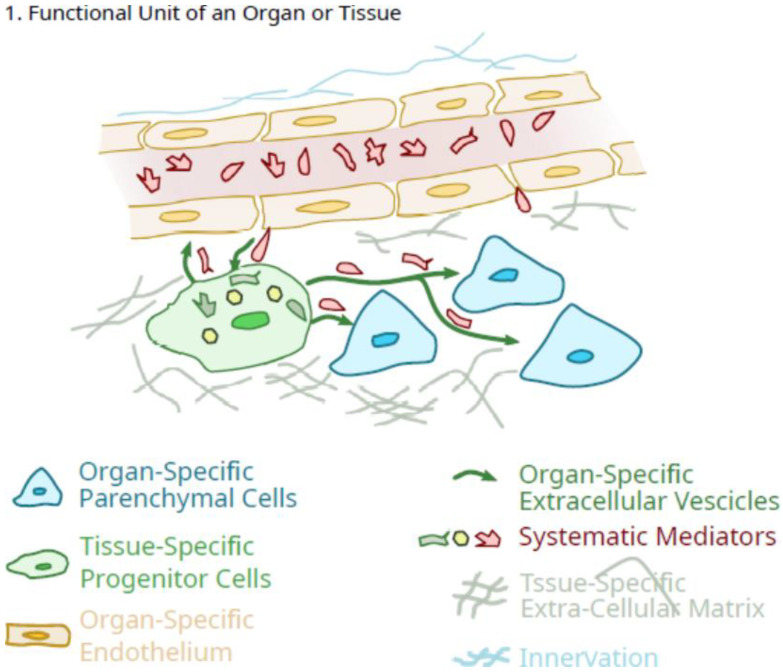
Schematic of the role of tissue-specific pluripotent cells in the integrated function unit of an organ.

**Figure 2 ijms-23-05496-f002:**
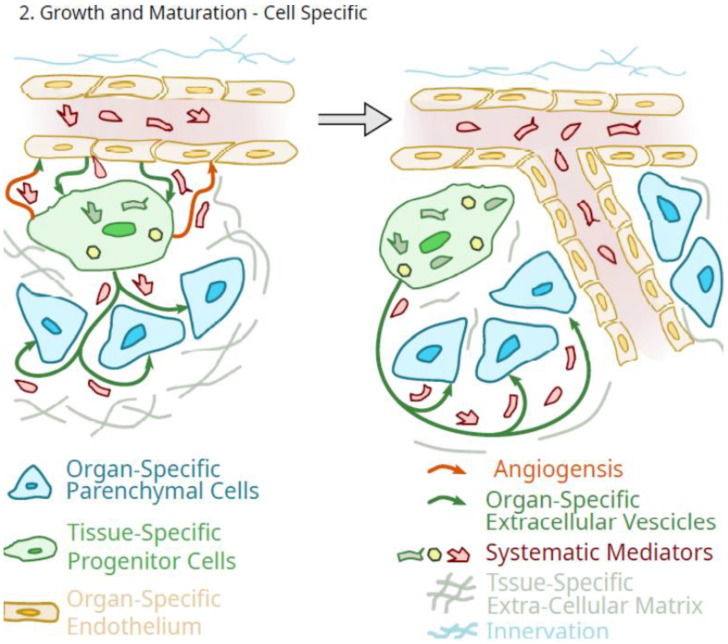
Schematic of the potential role of tissue-specific pluripotent cells in the regulated growth of an organ.

**Figure 3 ijms-23-05496-f003:**
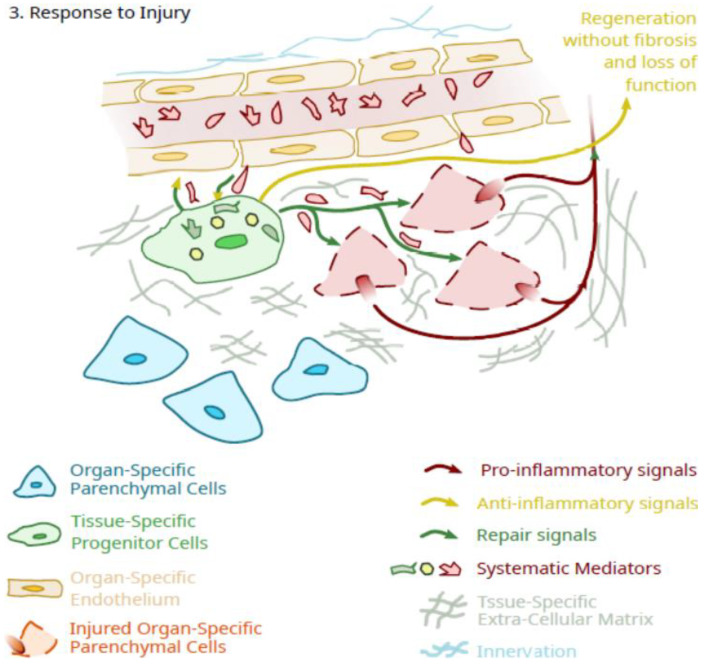
Schematic of the potential role of tissue-specific pluripotent cells in facilitating tissue repair after an injury to parenchymal cells of the tissue.

## Data Availability

This review does not contain any original data and therefore, there is no data to deposit.

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
