# Peer review of "One of the Primary Functions of Tissue-Resident Pluripotent Pericytes Cells May Be to Regulate Normal Organ Growth and Maturation: Implications for Attempts to Repair Tissues Later in Life"

_ijms, 2022, doi:10.3390/ijms23105496_

Round 1

Reviewer 1 Report

This review, which places tissue-specific MSCs at the center of regulating organ growth and maturation during development while also playing an important role in maintaining organ integrity and repair later in life, is interesting, well-written, and fairly comprehensive. 

However, I would suggest simplifying the many terms used in the introduction and discussion: MSC/PRMC/medical signaling cells versus mesenchymal stem cells or MSCs/pericytes, which makes it a bit confusing and difficult to read. The paper could focus more on the roles of these cells in the repair processes and maintenance of organ integrity during aging and injury. 

The paper is also very long. Some cuts in the introduction would be welcome. 

Author Response

Please see attached File regarding my responses to individual points raised.

Reviewer 2 Report

The review is thought-provoking and comprehensive.

Comments

  1. Regarding the discussing issues, it would be interesting the author’s opinion on the possible mechanisms for whole body control mechanism coordinating MSCs function in all organs in addition to their inter-organ regulating activity.
  2. Page 11, the last paragraph: As every tissue contains MSCs the effects of the young versus old blood can be explained by the activity of the growth factors [PMID: 16400016]. As for possible molecular mechanisms, the reprogramming of energy metabolism might help to override the old-age limitations [PMID: 28042296].

Author Response

Please see the attached file containing my responses to the individual points raised by Reviewer #2
